# A New Bit Repair Fast Reroute Mechanism for Smart Sensors IoT Network Infrastructure

**DOI:** 10.3390/s20185230

**Published:** 2020-09-14

**Authors:** Jozef Papan, Pavel Segec, Oleksandra Yeremenko, Ivana Bridova, Michal Hodon

**Affiliations:** 1Department of InfoCom Networks, University of Žilina, 010 26 Žilina, Slovakia; pavel.segec@fri.uniza.sk (P.S.); ivana.bridova@fri.uniza.sk (I.B.); 2Department of Infocommunication Engineering, Kharkiv National University of Radio Electronics, 61166 Kharkiv, Ukraine; oleksandra.yeremenko.ua@ieee.org; 3Department of Technical Cybernetics, University of Žilina, 010 26 Žilina, Slovakia; michal.hodon@fri.uniza.sk

**Keywords:** internet of things (IoT), Fast Reroute, bit repair (B-REP), failure repair

## Abstract

Today’s IP networks are experiencing a high increase in used and connected Internet of Things (IoT) devices and related deployed critical services. This puts increased demands on the reliability of underlayer transport networks. Therefore, modern networks must meet specific qualitative and quantitative parameters to satisfy customer service demands in line with the most common requirements of network fault tolerance and minimal packet loss. After a router or link failure within the transport network, the network convergence process begins. This process can take an unpredictable amount of time, usually depending on the size, the design of the network and the routing protocol used. Several solutions have been developed to address these issues, where one of which is the group of so-called Fast ReRoute (FRR) mechanisms. A general feature of these mechanisms is the fact that the resilience to network connectivity failures is addressed by calculating a pre-prepared alternative path. The path serves as a backup in the event of a network failure. This paper presents a new Bit Repair (B-REP) FRR mechanism that uses a special BIER header field (Bit-String) to explicitly indicate an alternative path used to route the packet. B-REP calculates an alternative path in advance as a majority of existing FRR solutions. The advantage of B-REP is the ability to define an alternative hop-by-hop path with full repair coverage throughout the network, where, unlike other solutions, we propose the use of a standardized solution for this purpose. The area of the B-REP application is communication networks working on the principle of packet switching, which use some link-state routing protocol. Therefore, B-REP can be successfully used in the IoT solutions especially in the field of ensuring communication from sensors in order to guarantee a minimum packet loss during data transmission.

## 1. Introduction

IoT architectures operate many types of different smart devices. The most used smart devices in the IoT are sensors that can be connected to the network using several technologies. In most cases, they are connected via wireless sensor networks (WSNs) [1,2,3,4]. With the growing popularity of the Internet of Things solutions, the number of WSN deployments is growing, as is the number of sensors connected in them, which of course leads to an increase in the amount of data generated by these sensors. Such devices are no longer used only for simple tasks, but also in comprehensive scenarios and services. Consequently, from the communication point of view, new challenges arise here. The whole communication chain must meet new requirements not only for simple data delivery but also must guarantee some reliability parameters of different data transfers such as availability, reliability, and network fault tolerance [5,6,7,8].

For example, if there is a connectivity loss or a change of network topology within the delivery chain, routing protocols of external base stations (BS) networks [9,10], in analogy to the Open Shortest Path Free (OSPF), respond to this change by re-converging the network [7,8,11,12,13]. During this process, routing protocols must update their routing information. Therefore, routers affected by the failure begin to send update messages to other routers that include recorded changes in the network topology. Receiving routers can then respond to topological changes and adjust their routing decisions. The time of the network convergence process depends mainly on the size and complexity of the network (number, density of nodes and links), as well as the routing protocol used. In a period of network convergence, network routers do not have valid routing information needed to deliver data properly. The loss, duplication, or other negative effects on data flows may significantly increase, and applications, services, or hosts may be unreachable, interrupted, or may provide unsatisfactory quality. This may be unacceptable for the quality of some services, such as critical or real-time. To address this issue and limit the impact of different network convergence times on the correct delivery of transmitted data, the FRR mechanisms have been developed [14,15,16,17].

The primary feature of many existing FRR mechanisms is the proactive calculation of alternative routes for each expected failure scenario. Alternative paths are calculated in advance on each individual router locally and before an unexpected specific network failure occurs. Once a connectivity failure is detected, the FRR mechanism of a local router quickly uses the pre-calculated alternative path to bypass a faulty connection. A simplified example is illustrated in Figure 1. Here, in the event of a connection failure of the link between the routers S and E, the S router uses a pre-prepared alternative path through the router N1 to bypass the failure and successfully deliver critical data during the convergence period. The alternative path is active and used at least until the network convergence process will complete. Thus, the main idea of FRR is that their recovery time of the affected communication is much faster than the convergence time of a routing protocol. This reduces the negative impact of failure on data delivery (loss, delay).

To further limit the effects of the error and to speed up the process of using the alternative route, FRR mechanisms may use other enhancement mechanisms, such as those for rapid detection of connection failure or neighbor unavailability (for example, bidirectional forwarding detection–BFD). These specialized mechanisms offer significantly faster detection of local failure than mechanisms built-in into current routing protocols (e.g., OSPF, IS-IS, EIGRP Hello mechanisms).

When using the FRR technology, the specific terms and definitions need to be introduced for denoting routers with unique FRR behavior [18,19,20,21,22]. The network topology shown in Figure 1 will be used for further explanation. The router that is detecting a link or neighboring router failure with the further activation of the FRR repair mechanism is defined as the source router (S). Consequently, router S becoming an active element of the FRR repair process (see Figure 1) referred to as the point of local repair (PLR). Router D denotes the destination router under discussion. Finally, the next-hop routers are specified by the routers N1, N2, N3, and others, which will compose the alternative path for FRR. The router not actively involved in the FRR repair process is designated the R router (here R1).

The paper presents a new bit repair (B-REP) FRR mechanism (hereinafter B-REP). The B-REP FRR mechanism is a type of proactive FRR mechanism. Unlike other FRR solutions, B-REP uses a standardized BIER header-based solution to circumvent failures. Mainly we focused on the ability to efficiently mark the entire alternative precalculated backup path thanks to its bit-string field. In addition, compared to other existing solutions (such as LFA or R-LFA), the B-REP mechanism can provide full repair coverage and repair all possible network failures within the topology. The B-REP can be deployed in an IP network that uses a link-state protocol, such as for example OSPF or intermediate system to intermediate system (IS-IS).

The B-REP is applicable to any network that operates the IP protocol stack and works with a link-state routing protocol. However, the deployment of the B-REP mechanism is more suitable for networks that are less dynamic, than usual WSN networks. B-REP works well for networks with static network nodes (some IoT deployments), or those networks behind the WSN BS towards the access/transport/ISP networks. In this environment, the B-REP mechanism provides the protection of important data flows generated from IoT/WSN devices–sensors and other devices. Our B-REP solution addresses the negative impacts of network failures and provides the solution that for the period of convergence guarantees correct packet delivery.

The remainder of this paper is structured as follows: Section 2 contains a summary of the latest knowledge in the FRR field and provides an analysis of existing solutions and their problem areas are discussed. Section 3 proposes the new B-REP FRR mechanism. Section 4 focuses on the evaluation of the B-REP mechanism and compares its features with other FRR solutions. Section 5 presents the conclusions of our work and plans for future research.

## 2. Related Works

Several existing solutions dealing with rerouting have been proposed in the IoT and IP network area. Reference [23] presents a new approach of jamming attack tolerant routing using multiple routes constructed on specific zones. This method separates the network into a number of zones and routes the candidate forward nodes of neighbor zones. When a system detects an attack, detour nodes in the network determine zones for proper rerouting and reroute packets intended for victim nodes through forwarding nodes in the obtained specific zones.

The authors of [24] present a comprehensive review of IoT sensing applications in WSNs, as well as the problems and tasks that need to be overcome. Some of the problem areas identified by authors are, for example, fault tolerance, the efficiency of the energy consumption, transmission interference, cost feasibility, and proper integration of these components.

Based on the identified issues and on our analysis of the IP FRR mechanisms properties, we can currently divide them into two generic groups/types. The first category of IP FRR mechanisms includes all proactive FRR mechanisms. The second category contains a family of reactive FRR mechanisms.

A characteristic of proactive FRR mechanisms is that these mechanisms calculate the alternative path in advance before a failure occurs. They act proactively when they already have alternative paths ready to respond to an error, depending on the location of the error. In the event of a failure, this backup path is then immediately installed into the routing table and used by a routing engine to redirect traffic around the failure. The interruption of operation caused by a network failure is therefore only for the time necessary to install a pre-prepared alternative route. This is considered as their undeniable advantage [25]. However, mechanisms in this category also have certain disadvantages identified. These include such properties as the pre-computation, the dependence on link-state routing protocols, complexity of internal algorithm [21,26,27,28,29].

One of the first mechanisms developed in the field of fast network recovery is the loop-free alternates (LFA for short) mechanism. LFA uses alternative next-hop neighbors that are able to deliver data without creating a routing loop. The selection of a suitable LFA candidate is ensured by several mathematical conditions. The basic version of the LFA mechanism [21,26,27] is suitable for certain topologies that are characterized by a high number of redundant links and suitable routing metrics. For topologies that do not meet these parameters, an extension of the classical LFA has been proposed, which is called the remote LFA (RLFA) [28,29]. RLFA has been proposed for situations where, for some reason, it is not possible to find suitable next-hop candidates according to the LFA conditions. RLFA uses tunneling mechanism to get around the failure towards remote tunnel end. Compared to the original LFA, the R-LFA provides higher repair coverage.

Other existing proactive FRR mechanisms are built on different approaches. The equal-cost multi-path (ECMP) [26,28] uses multiple routing paths that have the same metric in parallel. Multiple routing configurations (MRC) [28,30,31] uses different routing tables, while not-via addresses mechanisms [32,33] use specific addresses for explicit identification of failure. Furthermore, there are several FRR mechanisms exist based on alternative trees [19,34,35]. Amongst them, the maximally redundant trees (MRT) one is the most used [36,37].

The behavior of the reactive mechanism is different in that the alternative path is not calculated in advance. The backup path is constructed as an immediate reaction to the failure of the link itself and the protected flow transmitting through this link. Reactive FRR solutions do not calculate an alternative path in advance. However, the backup path is created after failure detection.

Reactive FRR mechanisms are not very numerous. They include the innovative multicast repair (M-REP) mechanism [36] and its enhanced version EM-REP [38]. The M-REP FRR mechanism uses the multicast [39,40,41] routing protocol—protocol independent multicast—dense mode (PIM DM) to flood traffic around failed element and create alternative FRR path. Enhanced version EM-REP adds support for the repair of multiple failures and link-state routing protocol such as OSPF or IS-IS.

The mechanisms mentioned so far have focused on addressing failure protection and convergence routing issues within the autonomous system (AS). Another FRR approach is represented by a mechanism called SWIFT. SWIFT is an FRR mechanism that is designed for the border gateway protocol (BGP), a protocol aimed at use for inter-AS routing. This mechanism is based on two approaches. First, a BGP router runs the SWIFT deduction algorithm that locates the failure and then tries to predict all prefixes that will be affected by the failure. The algorithm starts just after a router receives the routing update that contains first withdraw route information (prefix). Based on the calculations of this deduction algorithm, the router redirects the traffic to potentially affected prefixes to alternative routes that are not influenced by the failure (Figure 2).

In the case of BGP, the failure affects many prefixes at once. Therefore, as the second approach, the SWIFT introduces a new data plane coding scheme that allows the routing records concerned to be updated in a quick and flexible manner.

With the advent of relatively new approaches such as software defined networking (SDN), the FRR is emerging and being addressed in these application domains as well. The ability to calculate an alternate path in SDN depends on the option whether the SDN controller has access to a device imminently affected by a failure. As well as it depends on the round-trip time required for the communication between the SDN controller and devices influenced by the failure. The area of SDN is very diverse with a lot of proprietary approaches. The OpenFlow protocol is generally considered to be the most common standardized abstraction of the SDN internal communication. OpenFlow is used to install the data-match-action forwarding decisions applied through flow tables [42]. Therefore, as the link/node fault protection techniques there is the OpenFlow fast-failover [43]. This technique only operates in situations where the local node that detected the failure knows the alternative path. Unfortunately, such an alternative route may not always be available. In that case, the intervention of an SDN controller is required. However, as the controller is deployed remotely it may not have access to the node that detected the failure. If the controller is aware of the failure and the local node does not have an alternative path, the controller will setup the redirection at another SDN node in the network.

One possible solution is to implement a stateful mechanism for determining the condition of links and other rules regarding alternative routes in each individual SDN node. There are several OpenFlow extensions exist that address this issue. The following solutions can be mentioned here: OpenState [44], FAST [44], and SPIDER [42]. For example, SPIDER is a neighbor detection mechanism inspired by known techniques such as bidirectional forwarding detection (BFD) and multiprotocol label switching (MPLS) FRR. Unlike other solutions that are based on OpenFlow, detection and redirection in SPIDER are implemented exclusively in the data plane without the need to rely on a slower control plane [42]. This mechanism contains four basic methods:Local failover, a specific node directly detects a failure and reroutes traffic to the alternative path.Remote failover (Figure 3), a specific node (Figure 3, node 2) receives information from another node about the failure (Figure 3, Tag = F) and reroutes traffic to the alternative path (Figure 3, node 5).Node testing (heartbeat request/reply), used to verify node availability.Path probing, nodes periodically generate packets to check the reachability of the node or the route.

An overview of the basic parameters of existing FRR solutions as the output of our research in this area [18,38,45,46,47] is summarized in Table 1. In this table, we compared the most important properties of existing FRR mechanisms such as repair coverage, proactive behavior, dependency on link-state routing protocols, and prediction.

### 2.1. Problem Areas

According to the analysis given in the previous section, we have identified several FRR problems that can be classified into three basic areas which are described in the following subsections.

#### 2.1.1. Full Repair Coverage

According to the state-of-the-art analysis, we can state that some existing FRR mechanisms cannot provide full repair coverage. In other words, not all FRR mechanisms are able to repair and construct alternative backup paths for all possible failure scenarios. Furthermore, we can state that with the increasing repair coverage in most cases the complexity of the internal FRR algorithm also grows [48]. This may be an issue for complex and frequently changing network environment where routers with limited computation resources are installed, or low-priority FRR processes are used [48]. Therefore, one of the problem areas of current FRR solutions that provides full repair coverage is the complexity of an internal algorithm.

#### 2.1.2. Custom Alternative Path and Cost-Based Calculations

In some situations, a network administrator should also have the possibility to manually specify an alternative path. The administrator can define a custom alternative path that can avoid a group of routers potentially affected by the failure.

Existing FRR solutions such as LFA [33], R-LFA [16], TI-LFA [49] calculate alternative FRR paths according to the link metrics of used routing protocols. The calculation and construction of alternative paths must usually meet specific algorithm conditions. Only paths that satisfy them can be then selected as valid alternative routes. In certain topological situations, problems may arise where an alternative path exists, although it is possible that based on its metrics it does not meet mathematical conditions of an algorithm and cannot be used as an alternative path. Therefore, some existing paths are unnecessarily excluded under the given conditions. The alternative path is defined in each of the existing FRR mechanisms differently (tunneling, adding specific bits in the IP header).

#### 2.1.3. Research Goal (Research Aims and Objectives)

In previous subsections, we have identified some of the limitations of current FRR solutions that can be addressed and therefore offer new solutions that overcome them. Our work focused on the development of a new FRR mechanism concerned with the identified issues. Then the main contribution of this work is the proposal of the so-called Bit Repair (B-REP) algorithm. The B-REP algorithm is designed to repair of all possible failures within the specific network (100% repair coverage) with low calculation complexity.

The B-REP algorithm allows calculating an alternative path using standard link metrics or ignoring them, which means calculation without considering the metric’s limitations. Accordingly, if a path exists, it will be found and used. At the same time with the B-REP mechanism, an administrator can define manually a custom alternative path.

Several FRR mechanisms use various proprietary solutions to define an alternative route and to transit mechanisms related data. We opted for a standardized solution, namely multicast protocol–bit index explicit replication (BIER) [50]. This protocol uses a standardized BIER header (Figure 4) that contains a special field called bit-string (B-S, Figure 5). That allows us to use a header and its fields to define an alternative path as well as to transfer user data.

Bit-string is an array of bits, in which each bit indicates exactly one specific router in the BIER domain [52,53]. In other words, bit-string is an array data structure that efficiently stores router related information. The bits are arranged from the least significant bit (LSB) to the most significant bit (MSB). The use of bit-string in the FRR area will allow defining effectively an alternative backup path in the event of a link or node failure. The B-S value with the specified bits represents the routers through which the packet will be routed in the event of an error. It was the idea of using bit-strings that inspired us to develop the B-REP mechanism. A similar idea was used by greedy algorithms designed for fault-tolerant multicast delivery in the hypercube [54,55,56]. The authors suggested using a bit address to select the optimal hop-by-ho routing.

## 3. The Proposal of the B-REP FRR Mechanism

In this section, we provide a more detailed description of the B-REP mechanism operation principles. As mentioned before, the B-REP belongs to FRR mechanisms. However, it is designed to protect specific unicast flows delivered from customer devices or IoT sensors over the transport network. This network comprising the devices responsible for data delivery (i.e., layer 3 routers) is typically organized into administrative or policy areas or domains. For the proper operation of the B-REP mechanism, two new parameters and one mandatory condition are introduced.

The first necessary parameter is the device or router identifier (ID). We require that each router in such a common network domain must have a unique identifier. This means that if a given router is assigned a certain ID, no other router can use it within the domain. Therefore, we propose a new router ID, which we will call the B-REP router-ID (B-REP R-ID). The R-ID is similar to, for example, the BFR-ID in BIER enabled networks [50] and is crucial for the proper operation of the B-REP. The assignment process is not precisely specified. B-REP R-ID can be set manually (preferred option) by the administrator for example, or it can be created and derived from another unique identifier that is already assigned to the router. The second parameter is the bit-string (B-S) that is the special variable length array where the B-REP IDs of B-REP enabled routers are defined. Thereby de facto the whole alternative transport path can be defined. Being able to do this, we finally assume, as a prerequisite for the proper functioning of B-REP, that some type of link-state (LS) routing protocol is enabled and is running in the transport network. Link-state routing protocols provide all area routers with accurate topological information about all other LS routers in the area and therefore allow B-REP to specify precise bit string parameter as a definition of alternative paths applicable to different failure scenarios. B-REP requires access to the LSDB database of the given LS protocol. The obtained data are then processed by B-REP and the Dijkstra algorithm is used to calculate the alternative route. Dijkstra can run as a specific low-priority process when the router’s central processing unit (CPU) is idle. Therefore, the B-REP mechanism also calculates an alternative path during the CPU idle time. The alternative B-REP route is then stored in the B-REP backup path (B-REP BP) table (Table 2).

Let us show the use of these parameters on the example of a network running the OSPF routing protocol (Figure 6). OSPF is a LS routing protocol. In OSPF all active routers must be assigned unique router-IDs (R-ID). OSPF R-ID can be set manually or automatically. In the case of automatic assignment, the OSPF R-ID is assigned based on one of the local IP addresses of the router. Therefore, as one of the alternatives, it allows setting the B-REP R-ID according to the OSPF R-ID, which perfectly meets the condition of uniqueness. The mapping process is not specified however there can be applied some kind of algorithmic mapping. An example of R-ID and corresponding B-REP ID mapping is illustrated in Table 3.

OSPF as an LS routing protocol that allows a router to obtain and maintain precise topological information about all other routers, their interconnections, and network links of the area. This information is identical on all routers, which store it locally in a database called the link-state database (LSDB) database. Because all routers have the same LSDB databases that contain all topological information, including all routers, each B-REP router will assign a unique B-REP R-ID to itself as well as to all other routers in the area. As the next step, the router specifies corresponding bit-string values for each area router. This information is stored, and operations are performed inside of the B-REP Table (Table 3). The table is created by each B-REP router when the B-REP is activated and finally it will be the same on each router. The table is constructed by sorting all routers of the given area according to their OSPF router IDs ascending. Then the B-REP algorithm assigns them unique B-REP IDs and positions in the bit-string (Table 3). Because the LSDB is the same on all routers, the final B-REP table will be the same too. In case of a network failure, the content of the B-REP table is frozen because it is used to construct an alternative reroute path. When the convergence of the OSPF protocol ends, the process of creating the B-REP table is performed again, i.e., each B-REP router refreshes its own B-REP table.

As we can see from the example, the router with the lowest OSPF Router ID is Router 1. Its OSPF router ID is 1.1.1.1, therefore, it gets B-REP R-ID of 1. That gives R1 the least significant bit (LSB) position in the bit-string (…00001, Table 3, Figure 6). The Router 2, which is the second with the OSPF Router-ID 2.2.2.2 gets B-REP R-ID of 2 and the second position in bit-string (…00010, Table 3, Figure 6).

The basic idea of FRR mechanisms is to calculate an alternative route to bypass the local router’s failure in advance. The B-REP mechanism maintains this idea, as it also preliminary calculates an alternative route. Calculating an alternate route beforehand ensures that the routing engine of the router is able to quickly install and use the alternative route in the event of an error. Each B-REP-enabled router calculates alternative paths based on the protected link. In FRR terminology, the protected link is a link against whose failure (or by the failure of a directly connected neighbor) the network wants to be preserved. By default, the protected link in B-REP is set manually by an administrator. Next, based on the settings, B-REP calculates alternative paths to all destinations of protected flows routed over the protected link in order to decrease the impact of the link failure. For the simulation purposes, we expect one flow and one destination.

An alternative route is then used in the event of a protected link or its neighboring router failure. The B-REP mechanism uses Dijkstra’s algorithm to calculate an alternative path for each given destination by setting the protected link to a metric with a value at infinity.

When the router detects a connection failure with the primary next-hop router for a specific destination, it becomes the S router. The destination of an alternate path in terms of reroute scheme is the D router. In the considered case, it is Router 3 (Figure 7). The path from the router S to the router D must bypass the local failure detected by router S.

In the case of a link failure and specific protected flow of the customer, the router S already has a pre-calculated alternative path that contains all routers on the path including the destination router D. Following the failure protection procedures, the router S encapsulates packets of the original unicast protected flow with a new BIER header.

The BIER header includes the bit-string (Figure 8) field (BS field), which contains the exact definition of the pre-calculated path along which the packet will be routed (Figure 6). In case of several failures on the existing B-REP repair path, we assume the following behavior. The router that detects the failure adds a new/modified bit-string value that specifies the new alternative path. This means that packets are not again re-encapsulated, but only their LS field is modified. However, this property is still under further investigation.

Using the example of topology and link metrics from the Figure 7, the alternative path from the source to the destination is constructed via Router 2, Router 4 to Router 3 (Figure 9). The source router S (Router 1) will start encapsulating the original packets of a protected flow with a new BIER header immediately after the failure is detected. The router inserts the bit-string value “…01110” into the BS field (Figure 9, mark 1) that specifies the alternative route via Routers 2, 4, and 3. That BS value indicates the alternative path for other routers. An example of B-REP encapsulated packet is illustrated.

When a specific router, namely Router 2, receives a packet with a BIER header, it checks the bit-string value, finds the associated bit with its B-REP R-ID, and sets it to 0. The new BS value on R2 in our case is “…01100”. This operation ensures that the router that received the B-REP FRR packet does not receive it again. Without this operation, the router could receive the same packet again resulting in a micro-loop. After this operation, the router will need to specify the next-hop router on the way to the destination. The router checks the bit-string, and if it finds that it has a directly connected neighbor according to the bit-string value, it then forwards the packet to the given router. For our example, Router 2 has directly connected Router 4 with the bit-string position 01000. Therefore, Router 2 sends the packet to Router 4 (Figure 9, mark 2). Router 4 repeats the process and sends the packet to Router 3 (Figure 9, mark 3).

The bit-string field with a value that contains only one bit set to 1 indicates to a router that it is the destination router D. Router D then sets the last bit to 0 and removes the BIER header, i.e., router decapsulated the original packet. During the decapsulation process, the modified packet is restored to its original state. Router D then routes the packet according to its unicast routing table.

In the presented example, such a packet is received by the router 3. Router 3 receives a packet with the last bit set in the bit-string field (mark 4, 00100, Figure 9). Router 3, therefore, knows that it becomes the destination router D (the last one). It removes the BIER header and routes the packet to its destination based on the content of its unicast routing table as usual.

For clarity, we can also describe the activities of the B-REP mechanism using the following descriptive diagrams of B-REP activities. Figure 10 shows the process of B-REP activation and error response. Figure 11 shows the packet processing used on each B-REP router.

## 4. Evaluation

In this section, we provide the evaluation of the proposed B-REP mechanism. In its verification, we used simulations performed in the OMNeT ++ simulator. The results obtained from simulations performed in the OMNeT++ simulator are presented. In addition, a comparison with other existing FRR mechanisms has been performed.

### 4.1. Simulation in Deterministic Simulator

The correctness of the proposed B-REP FRR algorithm has been verified through the means of simulations performed in the Objective Modular Network Testbed in C++ (OMNeT++) discrete event simulator [57]. For the implementation of the algorithm, we used the INET [58] Framework library. The INET library provides OSPF routing capabilities. We tested the accuracy of the algorithm in several scenarios that simulated numerous types of failures for different topologies consisting of different numbers of interconnected routers. In these scenarios, we focused mainly on examining the appropriate delivery of packets belonging to the protected flow. These packets have to be correctly delivered to its destination in the event of a single failure. In the following section, we present an example of one of the comprehensive testing scenarios.

The topology used in the scenario is shown in Figure 12. It consists of the matrix of seventeen routers and four hosts, which form the routing domain. As the unicast routing protocol, we used the OSPFv2 protocol deployed in a single area deployment model. For the purpose of the simulation, we generate a data flow originated from a host named the Source to the destination, the host H3. This flow of data represents a protected flow of user packets, a correct delivery of which in the event of a link failure is insured by the B-REP algorithm.

In a stable network situation, unicast packets are delivered from the source to H3 along the shortest path selected by the OSPF (represented in Figure 12 and Figure 13 by the red line). Next in this scenario, we simulate a network failure, which is represented by shutting down the R6 router (all R6 interfaces go down). We then focus on verifying how the B-REP algorithm calculates the alternative path and how it uses it to protect user data in the event of the network failure. 

We expect that based on the link metrics the second shortest path is selected as an alternative (in Figure 12 and Figure 13 represented by the green line), then the correct entries in the B-REP table of all routers are created. We also examined the correct addition of BIER header to packets of the protected flow affected by a failure, correct reroute to an alternative path, and routing based on the BIER header to the destination router. Finally, we checked if all packets of the protected flow are correctly delivered.

The description of the simulation scenario is as follows. In the beginning, we need to wait the 50 simulation seconds (sims) that are required to complete all of the OSPF unicast routing processes, i.e., the creation and synchronization of LSDBs and the calculation of unicast routing tables. Then, at the time of 56 sims, the Source host starts generating its data flow towards the host H3. We call this flow a protected flow. At the time of 60 sims, we simulate the failure on the primary routing path, where all interfaces of the router R6 are turned off. This is where the B-REP mechanism starts working and delivers packets of the protected flow using an alternative route. At the time 70 sims, we restore the router R6 to original state. Here the B-REP FRR stops working and routers use their converged unicast routing tables (Table 4).

In the simulation, the B-REP mechanism deactivates when the source router detects that the protected interface is UP. In a real-life scenario, the B-REP will be deactivated by hold-down timer that is set to specific period. The duration of this period should be long enough so that the network can again successfully complete the convergence process.

#### Algorithm Behavior and Simulation Outputs

At the beginning of the simulation, once all OSPF processes have been finished (<50 ms), routers will compile their B-REP tables. Therefore, each router reads own OSPF LSDB, sorts the routers according to router’s OSPF-ID, and assigns them unique B-REP R-ID.

Next, all B-REP enabled routers precalculate alternative paths for its protected interfaces and protected flow according to OSPF LSDB. These pre-calculated backup paths are stored in B-REP BP table.

At time 56 sims, the source host starts generating packets of the protected flow sent with the destination address of the host H3. The B-REP algorithm is designed to protect only specific customers’ flows, i.e., flows that are identified by their source and destination addresses. Addresses as identifiers of protected flows must be preconfigured on the routers. Packets are delivered from the source to H3 along the shortest route as is selected by OSPF. At the time 60 sims, the described error occurs as is mentioned above. Disabling all interfaces of the R6 router will cause all its neighbors to detect its unavailability and thus, the network change. The speed of detection depends on the mechanism that routers use to check neighbors. Rapid failure detection can be guaranteed, for example, by a standardized BFD mechanism.

Next, all affected routers (R1, R2, R3, R5, R7, R9, R10, and R11) through OSPF update messages will begin flooding their new OSPF link-state advertisement (LSA) updates and begin the network convergence process. However, this process can take an unpredictable amount of time. Therefore, when the router R1 detects that the output interface with its primary next-hop router is no longer available, it begins to use B-REP. The R1 becomes the B-REP source router S. Subsequently, R1 start encapsulating unicast packets of the protected flows with a BIER header, into which it inserts the Bit-String routing value of the alternative path. The bit-string contains information about the pre-computed alternative path that will be used to route packets around the detected failure. This alternative path goes via R5 → R10 → R15 → R16 and is represented with the BS value of 000000010100010010 (Figure 14, Table 5).

Afterward, the packet with a new BIER header is subsequently routed to the next-hop router R5. Take note that, if the packet has a B-REP header, packet is not routed directly through the routing engine, but it is processed and routed through the B-REP process.

The explanation of Figure 14 is as follows: 192.168.1.2–OSPF router ID, 10.0.0.14–OSPF router ID of next next-hop router for the B-REP routing, 000000010100010010–bit-string that is inserted to the packet with BIER header.

The R5 receives the packet and starts to analyze its bit-string value. Accordingly, the R5 router detects its next-hop router, which is the R10 router with the B-REP ID = 9 (Table 5) and OSPF R-ID = 10.0.0.137. Subsequently, R5 modifies the BS value of the packet by setting its bit to zero. Then R5 immediately selects an outgoing interface that leads to the R10 neighbor and forwards the packet towards R10 (Figure 15).

Routers R10 and R15 repeat the same process as the R5 router forwards a packet to their next-hops (which is R15, respectively R16). The behavior is changed on the R16 router as it is the destination router D. R16 receives the packet, analyses the Bit-string value, and recognizes that only one bit in the bit-string is set to 1. This indicates that R16 is a decapsulation router. Therefore, R16 removes the BIER header and restores the packet of the protected flow to its original format (Figure 16). The decapsulation router is the end of the B-REP alternative path.

The behavior described above can be observed on the output obtained from the OMNeT ++ console listed in Table 6. We may see that at the time 56 sims, the source host starts sending packets of the protected data flow intended to the host H3. Packets follow the shortest route via routers: R1 → R6 → R11 → R16 → H3 (Table 6, green lines). At 60 sims, we simulate the failure of the R6 router. R1 detects its unavailability using the BFD connection failure detection mechanism. Therefore, the R1 becomes the Source router S and starts using the B-REP pre-calculated alternative path which goes through R1 → R5 → R10 → R15 → R16 → R10 (Table 6, blue lines). At the same time, all R6 neighbors are starting the OSPF update process. Until the OSPF R1 process completes and will provide actualized routing information, R1 will insert the pre-calculated path into the bit-string field of packet’s header. According to obtained simulation results, packets were successfully delivered using the B-REP algorithm around the failure router (Table 6, blue lines). At 70 sims, the router R6 is restored (all interfaces go UP), which means the Source router detects reconnection with R6 and disables B-REP mechanism. Packets are routed via original route as before (Table 6, black lines).

### 4.2. Evaluation of the B-REP Mechanism

The B-REP algorithm implementation uses the SPF algorithm in conjunction with any type of LS routing protocol, although we used OSPF for the pilot implementation. The B-REP SPF algorithm is applied to calculate the alternative shortest path used in the event of a failure. The main advantage of the B-REP FRR mechanism is that the algorithm implements an efficient and standardized way to mark an alternative path using the B-S field.

In the B-REP we use the bit-string to exactly define hop-by-hop routing behavior, where due to B-S we can precisely define the whole alternative path of routers chain. This feature might be used for an administrator to manually configure the alternative route in the event of a need.

Possibility of the explicitly defined alternative path can not only define the backup path close to a failed element in the network but also across the whole area which statistically can also be damaged.

Compared to other existing FRR mechanisms, the B-REP mechanism brings the new approach of defining alternative path (bit-string) into the FRR area and provides advantages in comparison with other mechanisms such as custom alternative path, easy implementation into existing architecture because of bit-string and 100% repair coverage. A summary of the advantages and disadvantages of the B-REP mechanism is given in Table 7.

A more exact comparison of the selected features with other existing FRR solutions is presented in Table 8 below. In this table we compare possibility of repairing all failures within the network, precomputing, modification of packets and support of custom alternative path.

The biggest time-consuming operation during the FRR process is the detection of link or node failure. For these purposes, the bidirectional forwarding detection (BFD) protocol is used. The BFD protocol is standardized by IETF in RFC 5880. Usually, the detection of the failure by BFD protocol is less than 30 ms depending on the timer settings. Another part of rerouting is switching to an alternate FRR path.

Existing proactive FRR solutions calculate alternative path in advance. Therefore, the rerouting time of the specific FRR mechanism is minimal because the alternative path is prepared and switchovers to that path immediately. B-REP algorithm calculates alternative path in advance, therefore its speed of recovery after link or node failure depends only on the time of failure detection, as is characteristic of proactive mechanisms.

## 5. Conclusions

The paper presents the bit-repair (B-REP) FRR mechanism, which provides advanced fast reroute solutions for IoT and IP network infrastructures. B-REP uses a standardized BIER header with a special bit-string field. That allows us to use a standardized header and its fields to define an alternative path as well as to transfer user data. The bit-string, in addition, allows us to efficiently define an exact alternative FRR path, which can be calculated by the Dijkstra algorithm or even manually defined by the administrator.

Some existing mechanisms, such as LFA or remote LFA calculate an alternative path according to specific metric conditions. However, in topologies with inappropriate metrics, these mechanisms are not able to choose an alternative path, but our algorithm is. We also add the ability to ignore metric-based calculations in our proposal, allowing us to select any possible physical alternative path. This mechanism can provide link or node protection and is suitable for any link-state protocols. These properties of the B-REP algorithm make it possible to achieve full repair coverage, which provides the protection against all possible failures in the network if a physically alternative path is presented.

The B-REP mechanism, as a proactive FRR mechanism, shares with other FRR mechanisms the properties of this family, which are identified as limitations. This includes CPU consumption during preliminary calculations and memory consumption for storing them. With FRR mechanisms that create a remote alternate path, B-REP uses the encapsulation of the original packets, which of course increases the overhead of the transmitted data. However, this limitation is partially addressed by using the Bit-String to define the entire transmission path. The speed of existing proactive FRR mechanisms including B-REP is similar, but the differences are in the way how they calculate alternative FRR path, how effective the results are and how the alternative path is constructed. According to our research, there is no FRR mechanism of all solutions. All of them have some advantages and disadvantages and B-REP brings his perspective on the solution to the issue.

The B-REP mechanism was fully implemented, and its correctness was tested using the OMNeT++ discrete event simulator. We have performed various extensive tests of the implementation in different network topologies, which validated the functional correctness of all B-REP sides. The use of Bit-String is unique, and it is possible to apply it in WSN networks, IoT design, and other areas as well, which will be studied in future work. Besides that, our future work will focus on the investigation of other related B-REP research issues, such as the addressing of multiple error occurrence, B-REP resource demands, and the B-REP bit-string-based source routing. We are also preparing the implementation of several existing FRR mechanisms into the OMNeT++ simulation tool, which will allow a better comparison of existing solutions and bring new knowledge in the field.

## Figures and Tables

**Figure 1 sensors-20-05230-f001:**
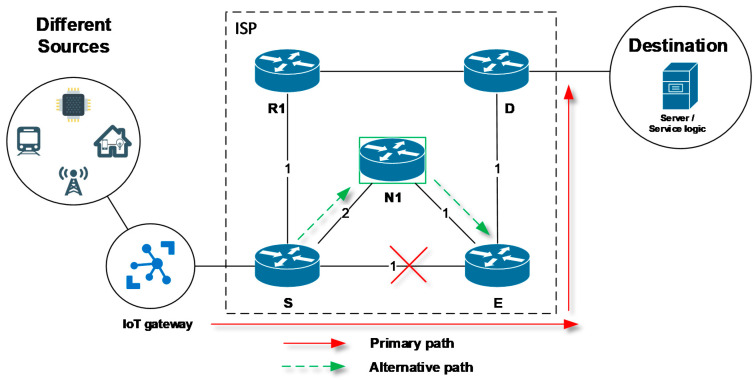
An example of the Fast ReRoute protection.

**Figure 2 sensors-20-05230-f002:**
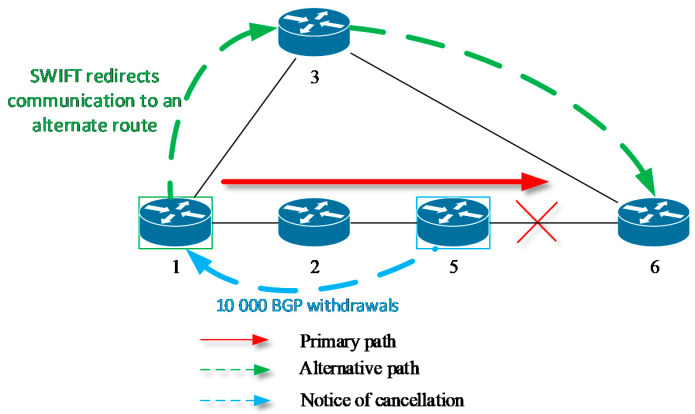
Principle of FRR mechanism SWIFT for BGP protocol.

**Figure 3 sensors-20-05230-f003:**
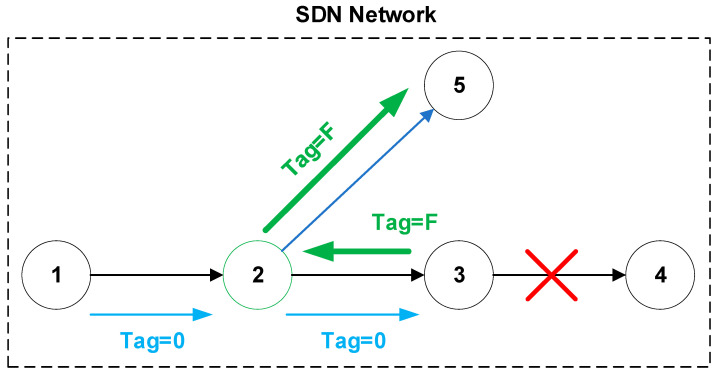
SPIDER—remote failover method.

**Figure 4 sensors-20-05230-f004:**
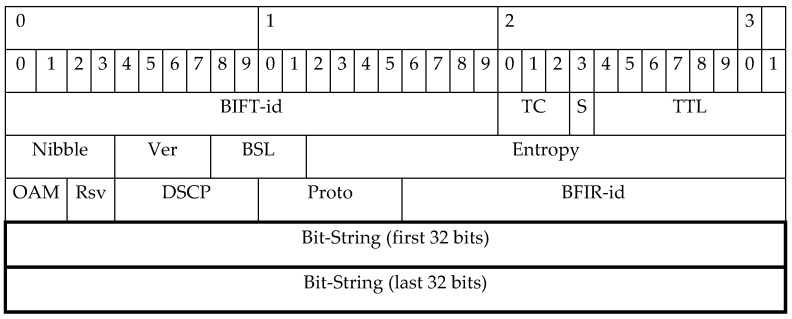
BIER header [51].

**Figure 5 sensors-20-05230-f005:**
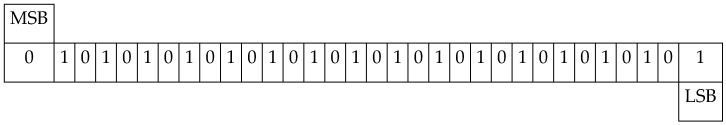
Bit-String.

**Figure 6 sensors-20-05230-f006:**
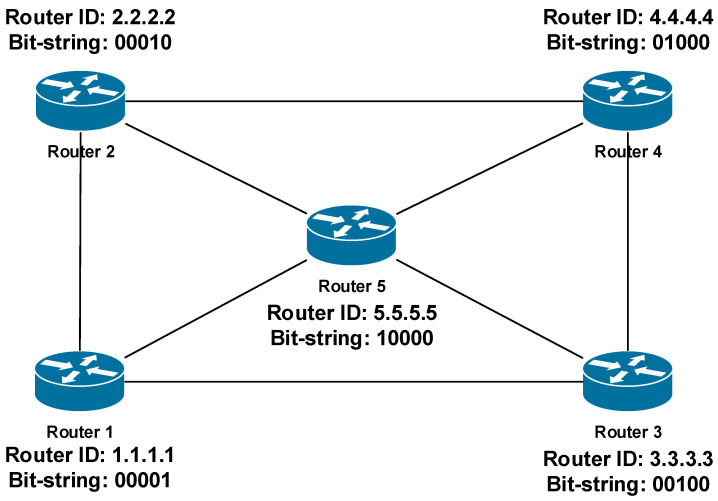
Allocation of Bit-Strings by the Router ID.

**Figure 7 sensors-20-05230-f007:**
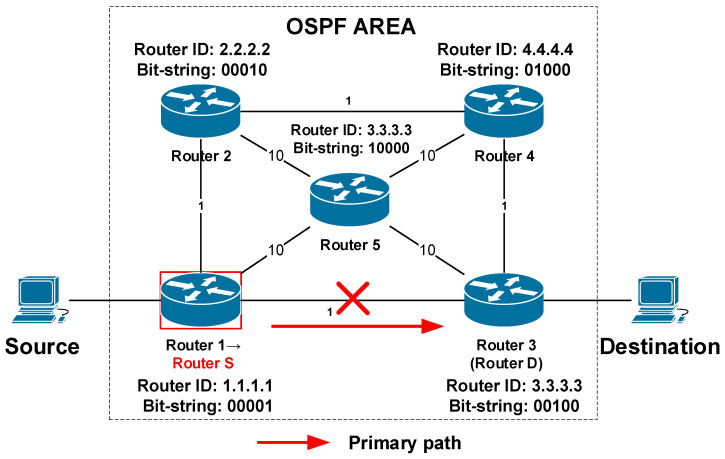
Detection of a failure and the reaction of the B-REP mechanism.

**Figure 8 sensors-20-05230-f008:**
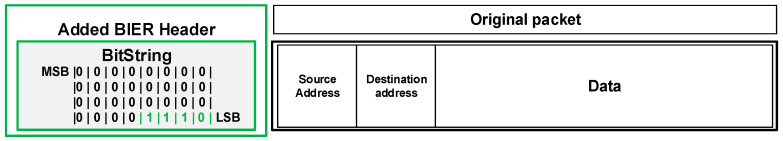
B-REP encapsulation of the IP packet.

**Figure 9 sensors-20-05230-f009:**
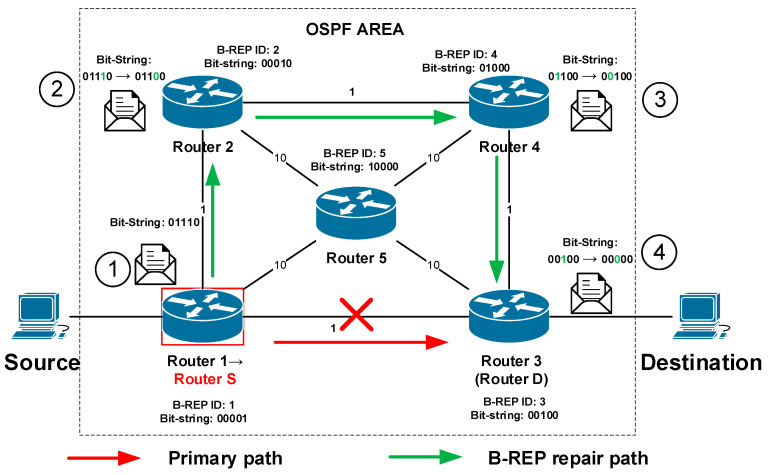
Example of a modified bit-string.

**Figure 10 sensors-20-05230-f010:**
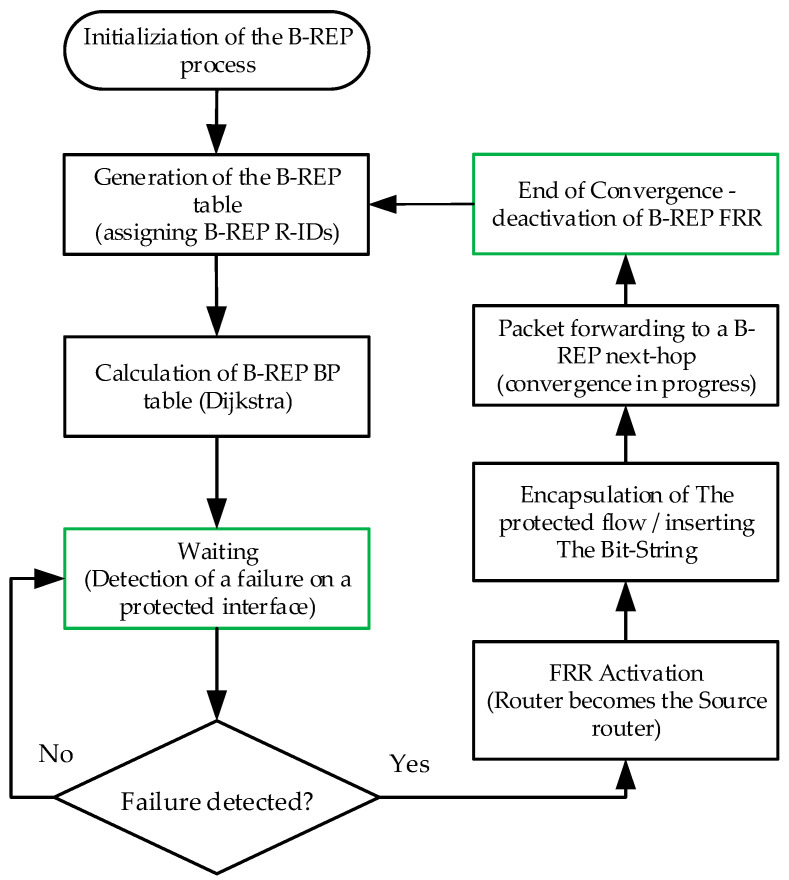
Diagram of B-REP activation and error response process.

**Figure 11 sensors-20-05230-f011:**
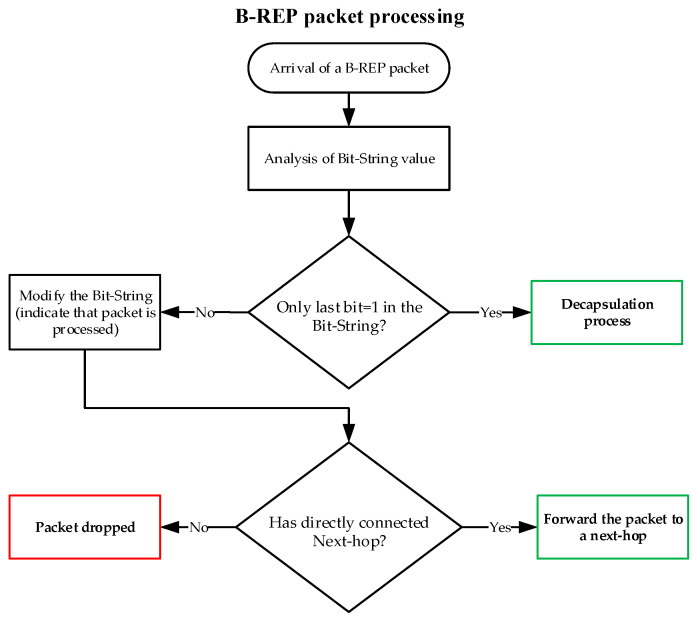
Processing of B-REP packet.

**Figure 12 sensors-20-05230-f012:**
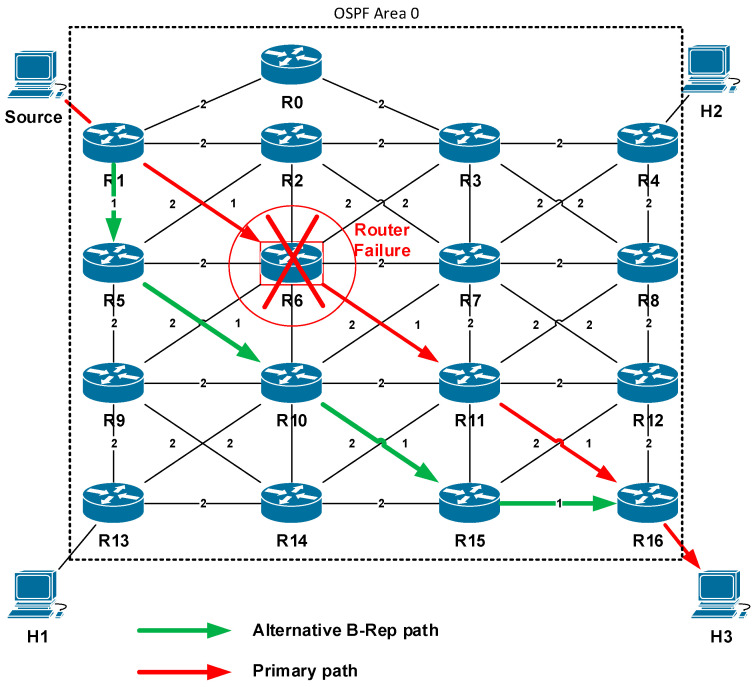
Simulation topology.

**Figure 13 sensors-20-05230-f013:**
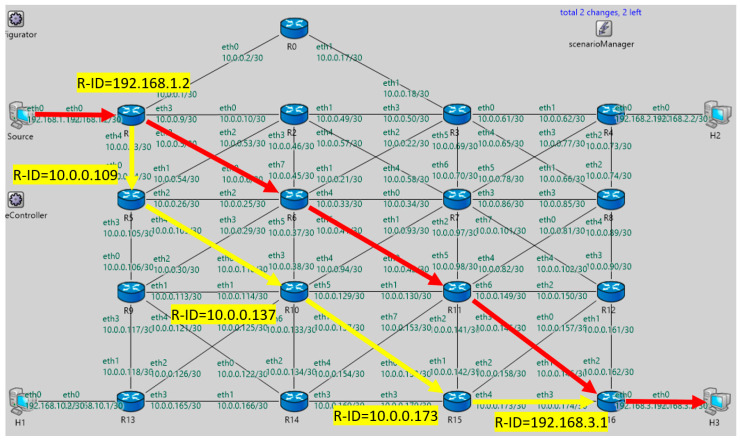
Visual output from the OMNeT++ simulation.

**Figure 14 sensors-20-05230-f014:**
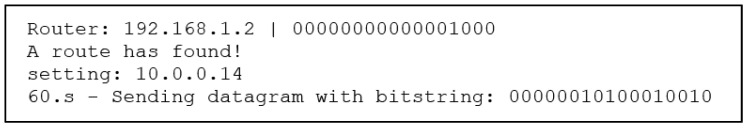
Inserting Bit-String (Source router).

**Figure 15 sensors-20-05230-f015:**
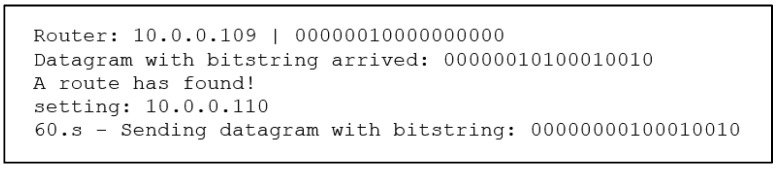
Processing of B-REP packet (R5).

**Figure 16 sensors-20-05230-f016:**
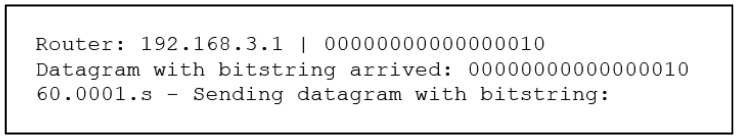
Output from the OMNeT++ console.

**Table 1 sensors-20-05230-t001:** Comparison of FRR solutions.

Solution	Full Repair Coverage	Proactive	Dependency on Routing Protocols	Prediction
		Behavior		
BIER-TE	Yes	Yes	No	No
Directed LFA	No	Yes	Yes	No
LFA	No	Yes	No	No
MPLS-TE	No	Yes	No	No
MRC	Yes	Yes	Yes	No
MRT	Yes	Yes	Yes	No
Not-Via	Yes	Yes	Yes	No
Remote LFA	No	Yes	Yes	No
TI-LFA	Yes	Yes	Yes	No
SWIFT (BGP)	Yes	Yes	Yes	Yes
M-REP	Yes	No	No	No
EM-REP	Yes	No	No	No

BIER-TE-Bit Index Explicit Replication-Traffic Engineering. LFA-Loop-Free Alternate. MRC-Multiple Routing Configurations. MRT-Maximally Redundant Trees. TI-LFA–Topology Independent Loop-Free Alternate.

**Table 2 sensors-20-05230-t002:** B-REP BP table.

Protected Interface	Destination B-REP R-ID (OSPF Router ID)	Bit-String Value
Interface R1-R3	3 (3.3.3.3)	…01110 (LSB)
Interface R1-R3	4 (4.4.4.4)	…01010 (LSB)
…	…	…

**Table 3 sensors-20-05230-t003:** B-REP Table—Bit allocation according to Bit-String.

Router	Router ID (OSPF)	B-REP R-ID	Bit-String Position (B-REP)
Router 1	1.1.1.1	1	…00001 (LSB)
Router 2	2.2.2.2	2	…00010
Router 3	3.3.3.3	3	…00100
Router 4	4.4.4.4	4	…01000
Router 5	5.5.5.5	5	…10000

**Table 4 sensors-20-05230-t004:** Description of the simulation scenario.

Time	Description of Action
<50	Time necessary for the OSPF convergence and stabilization of network processes.
56	Source host begins generating the flow
60	Router R6 failure
70	Restoration of R6

**Table 5 sensors-20-05230-t005:** B-REP table.

Bit-String	Name	OSPF R-ID	B-REP ID
000000000000000010	R16 (Destination router)	192.168.3.1	2
000000000000001000	R1 (Source router)	192.168.1.2	4
000000000000010000	R15	10.0.0.173	5
000000000100000000	R10	10.0.0.137	9
000000010000000000	R5	10.0.0.109	11

**Table 6 sensors-20-05230-t006:** Output from the OMNeT++ simulation.

Time	Source/Destination	Name	Destination Address	
0.079876921815	R12 --> R16	OSPF_HelloPacket	IPv4: 10.0.0.161 > 224.0.0.5	Network convergence
0.079977739817	R15 --> R16	OSPF_HelloPacket	IPv4: 10.0.0.173 > 224.0.0.5	
0.081024165641	R1 --> R6	OSPF_HelloPacket	IPv4: 10.0.0.5 > 224.0.0.5	
0.081275865377	R10 --> R13	OSPF_HelloPacket	IPv4: 10.0.0.125 > 224.0.0.5	
0.082012804148	R6 --> R9	OSPF_HelloPacket	IPv4: 10.0.0.29 > 224.0.0.5	
0.082307311152	R14 --> R10	OSPF_HelloPacket	IPv4: 10.0.0.134 > 224.0.0.5	
0.079876921815	R12 --> R16	OSPF_HelloPacket	IPv4: 10.0.0.161 > 224.0.0.5	
0.079977739817	R15 --> R16	OSPF_HelloPacket	IPv4: 10.0.0.173 > 224.0.0.5	
	… the output has been shortened …			
56.00006842	→R1	UDPBasicAppData-185	192.168.3.2	Network without errors
56.00008084	R01 → R06	UDPBasicAppData-185	192.168.3.2	
56.00009326	R06 → R11	UDPBasicAppData-185	192.168.3.2	
56.00010568	R11 → R16	UDPBasicAppData-185	192.168.3.2	
56.0001181	R16 → H3	UDPBasicAppData-185	192.168.3.2	
58.00007242	→R1	UDPBasicAppData-185	192.168.3.2	
58.00008484	R1 → R06	UDPBasicAppData-185	192.168.3.2	
58.00009726	R06 → R11	UDPBasicAppData-185	192.168.3.2	
58.00010968	R11 → R16	UDPBasicAppData-185	192.168.3.2	
58.0001221	R16 → H3	UDPBasicAppData-185	192.168.3.2	
	… the output has been shortened …			
60.00009726	→R1	UDPBasicAppData-B-REP	192.168.3.2	B-REP FAST REROUTE
60.00010968	R1 → R5	UDPBasicAppData-B-REP	192.168.3.2	
60.0001221	R5 → R10	UDPBasicAppData-B-REP	192.168.3.2	
60.00013452	R10 → R15	UDPBasicAppData-B-REP	192.168.3.2	
60.00014694	R15 → R16	UDPBasicAppData-B-REP	192.168.3.2	
60.00015936	R10→ H3	UDPBasicAppData-B-REP	192.168.3.2	
60.00017178	→R1	UDPBasicAppData-B-REP	192.168.3.2	
60.0001842	R1 → R5	UDPBasicAppData-B-REP	192.168.3.2	
60.00019662	R5 → R10	UDPBasicAppData-B-REP	192.168.3.2	
60.00020904	R10 → R15	UDPBasicAppData-B-REP	192.168.3.2	
60.00022146	R15 → R16	UDPBasicAppData-B-REP	192.168.3.2	
60.00023388	R16→ H3	UDPBasicAppData-B-REP	192.168.3.2	
	… the output has been shortened …			
70.00007242	→R1	UDPBasicAppData-185	192.168.3.2	Restoration
70.00008484	R01 → R06	UDPBasicAppData-185	192.168.3.2	
70.00009726	R06 → R11	UDPBasicAppData-185	192.168.3.2	
70.00010968	R11 → R16	UDPBasicAppData-185	192.168.3.2	
70.0001221	R16 → H3	UDPBasicAppData-185	192.168.3.2	

**Table 7 sensors-20-05230-t007:** Properties of the B-REP algorithm.

Advantages	Disadvantages
Suitable for networks of any size	Pre-computation
Applicable for a link-state routing protocol	Encapsulation packet overhead
100% repair coverage	
Possibility to define a custom path	
Relatively easy implementation (Bit-String)	

**Table 8 sensors-20-05230-t008:** Comparison of the B-REP mechanism with existing solutions.

Title	100% Repair Coverage	Custom Alternative Path	Precomputing	Packet Modification
B-REP	Yes	Yes	Yes	Yes
EM-REP	Yes	No	No	Yes
ECMP FRR	No	No	Yes	No
BIER-TE (M)	Yes	No	Yes	Yes
Directed LFA	Yes	Yes	Yes	Yes
LFA	No	No	Yes	No
MPLS-TE FRR	No	Yes	Yes	Yes
MRC	Yes	Yes	Yes	Yes
MRT	Yes	No	Yes	Yes
Not-Via Addresses	Yes	No	Yes	Yes
Remote LFA	No	No	Yes	Yes
TI-LFA	Yes	No	Yes	Yes

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
