# Peer review of "A New Bit Repair Fast Reroute Mechanism for Smart Sensors IoT Network Infrastructure"

_sensors, 2020, doi:10.3390/s20185230_

Round 1
Reviewer 1 Report
The well-known fact that the network and corresponding services fault-tolerance provided to end-users depend on its readiness for various failures that may occur due to multiple internal and external factors. Therefore, in the event of a malfunction of the network, including the IoT infrastructure, fault-tolerance facilities must be able to restore and provide an acceptable level of customer service. At the same time, in order to meet stringent reliability requirements, the network must be able to respond effectively to failures that become more likely with increasing network size. In addition, the most common network failures are links and nodes failures. It is known that the key technological tool to ensure fault-tolerance is routing protocols, to the level of efficiency of which there are high requirements for computational complexity and scalability of network solutions.
In recent years, the concept of software-defined networks (SDN) and its various types, including hybrid and global software-defined networks, have become very popular. This is due to the many advantages of SDN and Network Function Virtualization (NFV) over traditional networks, especially in terms of performance, flexibility, management, and optimization of the network and its processes.
In accordance with the above mentioned, the article “New Bit Repair fast reroute mechanism for smart sensors IoT network infrastructure” is devoted to the relevant topic. The main purpose of the presented research is the development of the new Bit Repair (B-REP) Fast ReRoute (FRR) mechanism, the essence of which is using of a special BIER header field (Bit-String) in order to the explicit indication of the alternative (backup) for the flow of packets routing in a case of failure.
The main advantages of presented work:
1) Related work analysis and characteristics of the proactive and reactive Fast ReRoute mechanisms have been presented. Moreover, SDN-based solutions are shown in detail.
2) The B-REP FRR mechanism proposed and well explained using examples and schemes.
3) The evaluation section is present in the article, as well as simulation results in the OMNeT++ environment. The case of router failure is considered. Mechanism behavior and simulation outputs with the corresponding screenshots are presented.
4) In the subsection devoted to the evaluation of the B-REP FRR, its properties (pros and cons) have been analyzed in line with the comparison of the presented solution with existing ones.
While the disadvantages and suggestions are the following:
1) In Figure 11, the text near routers is hardly readable. The figure should be improved or the necessary explanations provided.
2) Figures 12-14 need more explanation also.
3) The schematic presentation and state diagrams of the proposed algorithm are not provided, which could be helpful in further investigations.
4) Future research should be provided.
Taking all into account, the mentioned disadvantages do not reduce the practical importance of the presented B-REP FRR mechanism in IoTinfrastructures, as well as its investigation.
Author Response
Dear reviewer, please see attachment.

Reviewer 2 Report
Comments:
- In Abstract, it is not clear what problem to solve. In Introduction, the authors didn’t clearly present the innovation of this paper.
- What if a new fail happens when a packet has already been wrapped with a BIER header? Can another BIER header be added to it? Or the header can be modified? The author didn’t consider it.
- “Following the failure protection procedures, the router S encapsulates packets of the original unicast protected flow with a new BIER header. The BIER header includes the Bit-String field (BS field), which contains the exact definition of the pre-calculated path along which the packet will be routed”. The authors try to encapsulate all the routers on tht new path into the BIER header, which seems not necessary. If that works, why doesn’t the authors use the BIER header once a flow occurs? In that way, the original routing can be replaced.
Besides, “router S already has a pre-calculated alternative path”. Is the path resuable? Since the alternative path has been found, why is the BIER header nededed?
- In Section 4, the authors just give only one example, which cannot illustrate the performance, nor the comparison with other FRR solution.
- In fact, there are greedy algorithms proposed for fault –tolerant multicast in hypercube. The authors use bit address to select optimal routing for each step. The contribution can be regarded as an application of it.
Author Response
Dear reviewer, please see attachment.

Reviewer 3 Report
In this paper, authors propose a new re-routing protocol relying on pre-computation. This new protocol overcomes two current limitations: providing full-repair coverage and enabling administrators to specify constraints on the backup routes.
The proposal is well presented with a lot of details and explanations.
My main remark is that I find the link between the proposal and the sensor networks not very well explained. Why the proposed protocol is adapted to this context?
Here are my remarks for each part:
1. Introduction
The introduction is perhaps the part of the paper that is the less clear
- You may describe the problem your solution is going to solve? Why your solution is better than the existing ones?
- What are the contributions of the paper?
2. Related works
The related works section is very complete. I am very happy to see this large variety of protocols from ecmp to mpls.
- It is not clear if the classification between proactive and reactive protocols is proposed by you and therefore is a contribution or if it is something from the literature. (the ambiguity comes from the sentence "we can currently further divide them into two generic groups/types.")
- All the cells of Table 1 should be detailed. For instance, it should be better to explain why LFA is full repair coverage, why it has no prediction.
- The sections 2.1 should be a full section. It may be a good thing to add some examples and to explain for each existing solution, why they do not meet these properties. It may also be a good thing to explain why these properties are essential in a WSN network.
3. The proposal
This part is very complete and easy to follow and to understand.
- It may be interesting to discuss what happen in case
- A sequence diagram of the exchanges between the routers could make the understanding easier.
- The way the administrator can add constraints on the backup path could be more explained because it is a key feature of your proposal.
4. Evaluation
The evaluation is correctly performed. However, I think it should be improved:
- What I think is missing in the evaluation is a comparison with the other protocols in terms of performance.
- An evaluation of the overhead could also be interesting. What is the overhead of the route pre-computation? The overhead of the BIER headers? Or the overhead of the size of the routing tables including the Bit-String position?
5. Conclusion
- It may be interesting to discuss the limitations of the proposed protocol a bit more. What is the impact on the performance? Why it is more adapted to a sensor network than a traditional one, etc.
- The future works should clearly be exposed.
Author Response
Dear reviewer, please see attachment.

Round 2
Reviewer 2 Report
No further comments.